# The Impact of Immunofunctional Phenotyping on the Malfunction of the Cancer Immunity Cycle in Breast Cancer

**DOI:** 10.3390/cancers13010110

**Published:** 2020-12-31

**Authors:** Takashi Takeshita, Toshihiko Torigoe, Li Yan, Jing Li Huang, Hiroko Yamashita, Kazuaki Takabe

**Affiliations:** 1Department of Breast Surgery, Hokkaido University Hospital, Hokkaido 060-8648, Japan; takashi.takeshita@huhp.hokudai.ac.jp (T.T.); hirokoy@huhp.hokudai.ac.jp (H.Y.); 2Department of Pathology, Sapporo Medical University School of Medicine, Hokkaido 060-8556, Japan; torigoe@sapmed.ac.jp; 3Department of Biostatistics and Bioinformatics, Roswell Park Comprehensive Cancer Center, Buffalo, NY 10001, USA; li.yan@roswellpark.org; 4Department of Surgical Oncology, Roswell Park Cancer Institute, Buffalo, NY 14263, USA; Jing.Huang@roswellpark.org; 5Department of Surgery, University at Buffalo Jacobs School of Medicine and Biomedical Sciences, the State University of New York, Buffalo, NY 14263, USA; 6Department of Breast Surgery and Oncology, Tokyo Medical University, Tokyo 160-8402, Japan; 7Department of Surgery, Yokohama City University, Yokohama 236-0004, Japan; 8Department of Surgery, Niigata University Graduate School of Medical and Dental Sciences, Niigata 951-8520, Japan; 9Department of Breast Surgery, Fukushima Medical University, Fukushima 960-1295, Japan

**Keywords:** breast cancer, the cancer immunity cycle, cancer genomics, tumor immune microenvironment, bioinformatics

## Abstract

**Simple Summary:**

The cancer-immunity cycle (CIC) is a series of self-sustaining stepwise events to fight cancer growth by the immune system. We hypothesized that immunofunctional phenotyping that represent the malfunction of the CIC is clinically relevant in breast cancer (BC) utilizing total of 2979 BC cases; 1075 from TCGA cohort, 1904 from METABRIC cohort were analyzed. The immunofunctional phenotype was classified as follows: hot T-cell infiltrated, high immune cytolytic activity (CYT), cold T-cell infiltrated, high frequency of CD8+ T cells and low CYT, and non-inflamed, low frequency of CD8+ T cells and low CYT. We demonstrated that immunofunctional phenotyping not only indicated the degree of anti-cancer immune dysfunction, but also served as a prognostic biomarker and HTI was inversely related to estrogen response.

**Abstract:**

The cancer-immunity cycle (CIC) is a series of self-sustaining stepwise events to fight cancer growth by the immune system. We hypothesized that immunofunctional phenotyping that represent the malfunction of the CIC is clinically relevant in breast cancer (BC). Total of 2979 BC cases; 1075 from TCGA cohort, 1904 from METABRIC cohort were analyzed. The immunofunctional phenotype was classified as follows: hot T-cell infiltrated (HTI), high immune cytolytic activity (CYT), Cold T-cell infiltrated (CTI), high frequency of CD8+ T cells and low CYT, and non-inflamed, low frequency of CD8+ T cells and low CYT. The analysis of tumor immune microenvironment in the immunofunctional phenotype revealed that not only immunostimulatory factors, but also immunosuppressive factors were significantly elevated and immunosuppressive cells were significantly decreased in HTI. Patients in HTI were significantly associated with better survival in whole cohort and patients in CTI were significantly associated with worse survival in triple negative. Furthers, HTI was inversely related to estrogen responsive signaling. We demonstrated that immunofunctional phenotype not only indicated the degree of anti-cancer immune dysfunction, but also served as a prognostic biomarker and HTI was inversely related to estrogen response.

## 1. Introduction

Breast cancer (BC) is the most commonly diagnosed cancer in women worldwide and the second leading cause of mortality due to cancer in women in the United States. The current lifetime risk of developing BC is one in every eight women [1]. Tumor infiltrating lymphocytes (TILs) are immune cells that have migrated to the tumor tissue and the local microenvironment [2]. TILs are classified into anti- and pro-cancer cells according to their action against cancer. CD8+ T cells, CD4+ Th1 cells, NK cells, B cells, classically activated macrophages (M1), and mature dendritic cells (DCs) contribute to anti-cancer functions. In contrast, CD4+ Th2 cells, regulatory B cells, CD4+ regulatory T cells (Tregs), myeloid-derived suppressor cells, and alternatively activated macrophages (M2) are pro-cancer immune cells [3]. TILs play the pivotal role in the initiation and progression of cancer [4]. The infiltration of TILs in breast tumors has been shown to affect survival and response to chemotherapy [2]. We have shown that the infiltration of TILs in primary breast tumors has a strong association with the timing of BC recurrence [5].

Recently, it has been revealed that the immune response in tumor tissue reflects a series of carefully regulated events that can be optimally addressed as a group, rather than individual cells [6]. The cancer-immunity cycle (CIC) is defined as a series of self-sustaining stepwise events required to gain efficient control of cancer growth by the immune system [6]. CIC has seven steps: release of cancer-cell antigens, cancer-antigen presentation, priming and activation of T cells, trafficking of T cells to tumors, infiltration of T cells into tumors, recognition of cancer cells by T cells, and killing of cancer cells. Some of the steps of CIC to modulate the existing activated anti-tumor T cell immune response have been found to function poorly in some cancer patients [7]. Depending upon which of the seven steps within the CIC failed, immune profiles can be grouped into the three immunohistological phenotypes; ‘Immune-desert’, ‘T-cell excluded’, and ‘T-cell inflamed’ [7]. The immune-desert phenotype occurs when the release of cancer-cell antigens, cancer-antigen presentation, and/or priming and activation of T cells are impaired. This phenotype is characterized by a paucity of T cells in the parenchyma or the stroma of the tumor. The T-cell excluded phenotype occurs when trafficking of T cells to tumors and infiltration of T cells into tumors are impaired. This phenotype is characterized by the presence of abundant immune cells that do not penetrate the parenchyma of these tumors, but are retained in the stroma surrounding the nest of tumor cells. The T-cell inflamed phenotype is characterized by the presence of abundant immune cells.

Although the above classification is focused on the location of the T-cells in cancer nests, it may be possible to classify based on the infiltration and activity of anti-cancer T cell response as follows: ‘non-inflamed’, ‘cold T-cell infiltrated’ and ‘hot T-cell infiltrated’. The ‘non-inflamed’ phenotype is characterized by both the immune-desert phenotype and the T-cell excluded phenotype, without T cells in cancer nests. The ‘cold T-cell infiltrated’ phenotype is characterized by the presence of abundant immune cells, where tumor-infiltrating T cell-mediated killing of cancer cells are impaired. The ‘hot T-cell infiltrated’ phenotype is characterized by the presence of abundant immune cells, where tumor-infiltrating T cell-mediated killing of cancer cells are functional.

However, it is difficult to clarify the immunofunctional phenotype, the CIC phenotyping classified based on the infiltration and activity of anti-cancer T cells in breast tumors, by immunohistochemistry, and it has not been fully understood. Recently, advances in the next-generation sequencing have made it possible to systematically explore the tumor immune microenvironment (TIME). In fact, researchers are developing some expression profile–based estimation of the abundance of specific cells in tumor microenvironment, using RNA sequencing of a bulk tumor tissue [8,9]. Further, it is well established that the immune cytolytic activity (CYT) scores represent anti-cancer immune activity and the killing of malignant cells by TILs [10].

We aimed to investigate how immunofunctional phenotype is related to TIME, gene expression profiles, and survival, utilizing collected data from The Cancer Genome Atlas (TCGA) and Molecular Taxonomy of Breast Cancer International Consortium (METABRIC) BC cohorts.

## 2. Results

### 2.1. Classification of Immune Phenotype Using CD8+ T Cells and CYT Score

The transcriptomic data and associated clinical parameters from 1075 women in the TCGA cohort and 1904 women in the METABRIC cohort were available for analysis. Correlation between the presence of CD8+ T cells and CYT is shown in Figure 1A. There was a weak correlation between the presence of CD8+ T cells and CYT in TCGA cohort (*r* = 0.518), but no correlation between them in METABRIC cohort (*r* = 0.379).

Based on the levels of CD8+ T cells and CYT, we classified TIME into following three categories (Appendix A). A group of patients who had high CYT was defined as hot T-cell infiltrated. A group of patients who had high frequency of CD8+ T cells and low CYT was defined as Cold T-cell infiltrated. Hot T-cell/Cold T-cell infiltrated corresponds to “T-cell inflamed” in immunohistological phenotype in the CIC [6,7]. A group of patients who had low frequency of CD8+ T cells and low CYT was defined as Non-inflamed. Non-inflamed corresponds to “Immune desert” and “T-cell excluded” in immunohistological phenotype in the CIC [6,7].

For this classification, we needed to define the CD8+ T cells and CYT cutoffs so that the presence of CD8+ T cells and CYT were the highest in hot T-cell infiltrated and they were lowest in non-inflamed in both cohorts. Thus, we defined the presence of CD8+ T cells of more than 85 th percentile of as high frequency of CD8+ T cells, which was the top 85.3% (917/1075) in TCGA cohort and the top 83.8% (1596/1904) in METABRIC cohort. We defined CYT of more than 40th percentile of as high CYT, which was the top 40.1% (431/1075) in the TCGA cohort and CYT of more than 8th percentile of as high CYT, which was the top 7.9% (150/1904) in METABRIC cohort. The box plots of the presence of CD8+ T cells and CYT compared between CIC phenotypes in TCGA BC cohort and METABRIC cohort are shown in Figure 1B,C.

### 2.2. Association of the Immunofunctional Phenotype with Clinical Features in Two Large BC Cohorts

We studied the relationship between clinical features of the primary tumor and the immunofunctional phenotype in TCGA BC cohort (Table 1) and METABRIC cohort (Table 2). In the TCGA BC cohort, patients in hot T-cell infiltrated were significantly associated with invasive lobular carcinoma (*p* < 0.0001), triple negative (TN) BC (*p* < 0.0001), and basal-like (*p* < 0.0001), compared with other groups. Patients in Cold T-cell infiltrated were significantly associated with older than 50 years (*p* = 0.0043), estrogen receptor (ER) positive (*p* < 0.0001), and progesterone receptor (PgR) positive (*p* = 0.011), compared with other groups. In METABRIC cohort, patients in hot T-cell infiltrated were significantly associated with higher tumor grade (*p* < 0.0001), ER negative (*p* < 0.0001), PgR negative (*p* < 0.0001), TN (*p* < 0.0001), and claudin-low (*p* < 0.0001), compared with other groups. These results indicate that hot T-cell infiltrated was related with higher tumor grade and was inversely related to hormone receptor (HR) positivity.

### 2.3. Hot T-Cell Infiltrated Has High Levels of Anti-Cancer Immune Cells and Low Levels of Pro-Cancerous Immune Cells; Non-Inflamed Has Low Levels of Anti-Cancer Immune Cells and High Levels of Pro-Cancerous Immune Cells

In order to evaluate the TIME in the immune phenotype, we analyzed the immune cell composition utilizing CIBERSORT in the TCGA BC cohort and METABRIC cohort (Figure 2). In pro-cancerous immune cells, M2 macrophages and Tregs were the lowest in the hot T-cell infiltrated phenotype and highest in the non-inflamed phenotype within the METABRIC cohort. In the TCGA BC cohort, M2 macrophages were the lowest in the hot T-cell infiltrated phenotype and highest in non-inflamed phenotype. However, Tregs was the highest in the hot T-cell infiltrated phenotype and lowest in the non-inflamed phenotype. Among anti-cancer immune cells, M1 macrophages, activated NK cells, T follicular helper cells, memory B cells, activated memory CD4+ T cells, and γδT cells were the lowest in the non-inflamed phenotype and highest in hot T-cell infiltrated phenotype in both the TCGA BC cohort and METABRIC cohort. However, activated DCs were the lowest in the hot T-cell infiltrated phenotype and highest in the non-inflamed phenotype in both the TCGA BC cohort and METABRIC cohort. These results indicate that the hot T-cell infiltrated phenotype had low levels of pro-cancerous immune cells and high levels of anti-cancer immune cells, whereas non-inflamed had high levels of pro-cancerous immune cells and low levels of anti-cancer immune cells.

### 2.4. Stimulatory and Inhibitory Factors of the CIC were Elevated in Hot T-Cell Infiltrated

Here, we explored the relationship between the immunofunctional phenotype and stimulatory or inhibitory factors in the CIC phenotype (Figure 3). In the analysis of stimulatory factors of release of cancer cell antigens in the CIC, the levels of tumor mutation burden (TMB) were the highest in the hot T-cell infiltrated phenotype. Interestingly, there was no significant difference in TMB between non-inflamed and cold T-cell infiltrated phenotypes. Non-silent/ silent mutation rate was higher in hot T-cell infiltrated, compared with cold T-cell infiltrated phenotype. Furthermore, the levels of single-nucleotide variant (SNV) neoantigens were the highest in the hot T-cell infiltrated phenotype. There was no significant difference in Indel neoantigens in among the immunofunctional phenotypes (Figure 3A). In the analysis of stimulatory factor of recognition of cancer cells in the CIC, HLA-I expression was the highest in the hot T-cell infiltrated phenotype. There was no significant difference between the non-inflamed and cold T-cell infiltrated phenotypes (Figure 3B). In the analysis of inhibitory factors of killing of cancer cells in the CIC, the expression of PD1, PD-L1, PD-L2, and IDO1 were the highest in the hot T-cell infiltrated phenotype. PD-L1 expression was the lowest in the non-inflamed phenotype, and PD-L2 expression was the lowest in the cold T-cell infiltrated phenotype (Figure 3C). These results indicate that stimulatory factors of release of cancer cell antigens and recognition of cancer cells and inhibitory factors of killing of cancer cells were significantly elevated in the hot T-cell infiltrated phenotype.

### 2.5. Gene Expression Profiles in the Immunofunctional Phenotype

In order to clarify the mechanisms associated with the immunofunctional phenotype, volcano plots and Gene Set Enrichment Analyses (GSEA) were performed in both the TCGA and METABRIC cohort. We show the correspondence of the immunofunctional phenotypes with the Hallmark gene sets in pre-ranked GSEA and volcano plots, which represent the distribution of the fold changes and adjusted *p*-values of 18,428 genes (Figure 4A). 12 mRNAs in the non-inflamed phenotype and 7 mRNAs in the cold T-cell infiltrated phenotype were up-regulated, compared with hot T-cell infiltrated, all of which were differentially expressed with fold change greater than log2 (1.5) and *p* < 0.05. In them, the expressions of *WHSC1L1*, *ASH2L*, and *DDHD2* were upregulated in both the non-inflamed and cold T-cell infiltrated phenotype, and *IL15RA* expression was upregulated in the hot T-cell infiltrated phenotype. On the other hand, in the cold T-cell infiltrated phenotype, 12 mRNAs were upregulated, compared with the non-inflamed phenotype. In the cold T-cell infiltrated phenotype, the expressions of *DMRTB1*, *MOG*, *KCNA4*, *ADAD1*, and *KLHL38* were upregulated, compared with other groups.

In pre-ranked GSEA, in both non-inflamed and cold T-cell infiltrated tumors, estrogen response early (non-inflamed; normalized enrichment score (NES) = 1.94, false discovery rate (FDR) *q* = 0.002, cold T-cell infiltrated; NES = 2.54, FDR *q* < 0.0001), estrogen response late (non-inflamed; NES = 1.76, FDR *q* = 0.011, cold T-cell infiltrated; NES = 2.17, FDR *q* < 0.0001) and oxidative phosphorylation (non-inflamed; NES = 1.78, FDR *q* = 0.011, cold T-cell infiltrated; NES = 2.29, FDR *q* < 0.0001) were enriched, compared with hot T-cell infiltrated tumors. In hot T–cell infiltrated tumors, interferon (IFN)-α/-γ response, tnfα signaling via nfκb, il6-jak-stat3, and il2-stat5 signaling were enriched, compared with non-inflamed or cold T-cell infiltrated tumors. On the other hand, in non-inflamed tumors, epithelial mesenchymal transition (NES = 2.89, FDR *q* < 0.0001), cell cycle related gene sets (G2M checkpoint; NES = 2.66, FDR *q* < 0.0001, E2F targets; NES = 2.33, FDR *q* < 0.0001, mitotic spindle; NES = 2.31, FDR *q* < 0.0001), and mTORC1 signaling (NES = 2.29, FDR *q* < 0.0001) were enriched, compared with cold T-cell infiltrated. In cold T-cell infiltrated tumors, estrogen response gene sets (early; NES = −1.96, FDR *q* = 0.004 and late; NES = 1.60, FDR *q* = 0.0024) were enriched, compared with non-inflamed tumors.

Figure 4B shows volcano plots, representing the distribution of the fold changes and adjusted *p*-values of 18,484 genes, and the Hallmark gene sets in pre-ranked GSEA, corresponding to the immunofunctional phenotype in the METABRIC cohort. In the comparative analysis between non-inflamed and hot T-cell infiltrated, 13 mRNAs were significant in Non-inflamed and 130 mRNAs were significant in hot T-cell infiltrated. In the comparative analysis between Cold T-cell infiltrated and hot T-cell infiltrated, 15 mRNAs were significant in non-inflamed and 78 mRNAs were significant in hot T-cell infiltrated, all of which were differentially expressed with fold change greater than log2 (1.5) and *p* < 0.05. The expressions of *TFF3, STC2, ESR1, AZGP1, EEF1A2, SCUBE2, TFF1, ANKRD30A, AGR3, MLPH, CA12,* and *FOXA1* were up-regulated in both Non-inflamed and Cold T-cell infiltrated. 77 mRNA expressions were commonly up-regulated in hot T-cell infiltrated. On the other hand, in the comparative analysis between non-inflamed and Cold T-cell infiltrated, there was no significant difference of mRNA expression.

In pre-ranked GSEA, in both non-inflamed and Cold T-cell infiltrated, estrogen response early (non-inflamed; NES = 2.45, FDR *q* < 0.0001, Cold T-cell infiltrated; NES = 2.49, FDR *q* < 0.0001) and estrogen response late (non-inflamed; NES = 2.08, FDR *q* < 0.0001, Cold T-cell infiltrated; NES = 2.02, FDR *q* < 0.0001) were enriched, compared with hot T-cell infiltrated. In hot T-cell infiltrated, IFN-α/-γ response, TNFα signaling via NFκB, IL6-JAK-STAT3, and IL2-STAT5 signaling were enriched, compared with non-inflamed or Cold T-cell infiltrated. On the other hand, in non-inflamed, cell cycle related gene sets (G2M checkpoint; NES = 1.84, FDR *q* = 0.002 and E2F targets; NES = 1.74, FDR *q* = 0.004) were enriched, compared with Cold T-cell infiltrated. In Cold T-cell infiltrated, IFN-α response (NES = −1.98, FDR *q* < 0.0001), IFN-γ response (NES = −2.68, FDR *q* < 0.0001), TNF-α signaling via NFκB (NES = −2.52, FDR *q* < 0.0001), IL6-JAK-STAT3 signaling (NES = −2.16, FDR *q* < 0.0001), IL2-STAT5 signaling (NES = −2.04, FDR *q* < 0.0001), and epithelial mesenchymal transition (NES = −2.19, FDR *q* < 0.0001) were enriched, compared with non-inflamed. These results indicate that, comparing hot T-cell infiltrated and non-inflamed or Cold T-cell infiltrated, early and late estrogen response were enriched in both Non-inflamed and Cold T-cell infiltrated. IFN-α/-γ response, TNFα signaling via NFκB, IL6-JAK-STAT3, and IL2-STAT5 signaling were enriched in hot T-cell infiltrated. Furthers, comparing non-inflamed and Cold T-cell infiltrated, cell cycle related gene sets (G2M checkpoint and E2F targets) were enriched in non-inflamed in both the TCGA BC cohort and METABRIC cohort. In Cold T-cell infiltrated, there were significant differences in gene signatures between cohorts. That is, estrogen response gene sets were enriched in the TCGA BC cohort, while IFN-α response, IFN-γ response, TNF-α signaling via NFκB, IL6-JAK-STAT3 signaling, IL2-STAT5 signaling, and epithelial mesenchymal transition were enriched in METABRIC cohort.

### 2.6. Hot T-Cell Infiltrated Was Associated with Better Prognosis in Two Large BC Cohorts

In order to verify that the immunofunctional phenotype can serve as a prognostic biomarker, we examined the relationship between the immune phenotype and prognosis in the whole cohort and subtypes, which were tested by the Kaplan–Meier method and verified by the log-rank test, in the TCGA BC cohort and METABRIC cohort (Figure 5A). In TCGA, patients with the Non-inflamed phenotype were significantly associated with worse overall survival (OS), and patients with the hot T-cell infiltrated phenotype were significantly associated with better OS in whole cohort (*p* = 0.01). The immune phenotype was not associated with OS in the HR+ human epidermal growth factor receptor 2 (HER2)-negative subtype. Patients with the non-inflamed phenotype were significantly associated with worse OS in the HER2-positive subtype (*p* < 0.001), and patients with the Cold T-cell infiltrated phenotype were significantly associated with worse OS in the TN subtype (*p* = 0.05). Patients with the hot T-cell infiltrated phenotype were marginally associated with better progression free survival (PFS) in whole cohort (*p* = 0.068) and the HER2-positive subtype (*p* = 0.008) (Appendix A). In METABRIC, patients with the hot T-cell phenotype infiltrated were marginally associated with better BCSS in whole cohort (*p* = 0.097), and patients with the Cold T-cell infiltrated phenotype were significantly associated with worse breast cancer specific survival (BCSS) in in TN subtype (*p* = 0.009) (Figure 5B). Furthermore, we examined the relationship between the immunofunctional phenotype and distant recurrence or local recurrence (Figure 5B and Appendix A). In distant recurrence analysis, patients with the hot T-cell infiltrated phenotype were significantly associated with better prognosis in whole cohort (*p* = 0.034) and in the TN subtype (*p* = 0.019). However, there were no significant differences between the immune phenotype and local recurrence.

## 3. Discussion

The immune response in cancer is a series of carefully regulated events that can best be addressed as a group rather than alone, and CIC is one of the rational models of the immune response to cancer [6]. We defined immunofunctional phenotype based on the number and CYT of tumor-infiltrating CD8+ T cells, and categorized it into ‘Non-inflamed’, ‘Cold T-cell infiltrated’, and ‘hot T-cell infiltrated’ phenotypes that represent the malfunction of CIC (Appendix A) [7]. We investigated how this immunofunctional phenotype categorization was related to TIME, gene expression profiles, and clinical outcomes utilizing collected data from TCGA and METABRIC cohorts. The hot T-cell infiltrated phenotype had high levels of anti-cancer immune cells and low levels of pro-cancerous immune cells, whereas the non-inflamed phenotype had low levels of anti-cancer immune cells and high levels of pro-cancerous immune cells (Figure 2). In addition, in pre-ranked GSEA, the hot T-cell infiltrated phenotype was significantly correlated with gene expression signatures of IFN-α/-γ response, TNFα signaling via NF-κB, IL6-JAK-STAT3, and IL2-STAT5 signaling, compared with the non-inflamed or Cold T-cell infiltrated phenotypes (Figure 4). From these results, this classification was closely associated with TIME.

This study generated three interesting results with clinical implications. First, the immunofunctional phenotype could identify the degree of anti-cancer immune dysfunction. We showed that elevated levels of immunostimulatory factors and immunosuppressive factors and decreased levels of immunosuppressive cells the hot T-cell infiltrated phenotype. In addition, we showed that PD-L1 expression was the lowest in the non-Inflamed phenotype and PD-L2 expression was the lowest in the Cold T-cell infiltrated phenotype (Figure 3). Anti-PD-L1/PD-1 therapy, one of the immune checkpoint inhibitors, has transformed the therapeutic landscape of a wide range of cancers [11] and has been successful in TNBC [12,13]. Various biological factors contribute to the effect of anti-PD-L1/PD-1 therapy, and it has been reported that immunohistological phenotype in the CIC was also closely related to the therapeutic effect. Unsurprisingly, immune desert and T-cell excluded, tumors rarely respond to anti-PD-L1/PD-1 therapy [7]. Interestingly, in T-cell excluded tumors, anti-PD-L1/PD-1 agents activate and proliferate stromal-associated T cells but have no clinical effect because they cannot invade tumors. In agreement, the generation or migration of tumor-specific T cells through the tumor stroma is the rate-limiting step in the CIC for this phenotype. In the T-cell inflamed phenotype, the clinical response to anti-PD-L1/PD-1 therapy is positive, but a response is uncommon in T-cell exhausted phenotype, where immune-cell infiltration is necessary but insufficient for inducing a response [7]. Xiao and colleagues also suggested that their classification of the microenvironment phenotypes would be a step toward personalized immunotherapy for patients with TNBC [14]. We showed that the proportion of the hot T-cell infiltrated phenotype was higher in the claudin-low subtype compared with other subgroups, suggesting that patients with this subtype may be especially responsive to anti-PD-L1 therapy (Appendix A). Thus, the immunofunctional phenotype categorization may allow the prediction of therapeutic strategies directed at individual immune biology to maximize the likelihood of a response to a particular treatment, such as anti-PD-L1/PD-1 therapy.

Second, immunofunctional phenotype served as a prognostic biomarker. We demonstrated that, in the TCGA BC cohort, patients with the non-inflamed phenotype were significantly associated with worse OS, and patients with the hot T-cell infiltrated phenotype were significantly associated with better OS in whole cohort (*p* = 0.01). In subtypes, patients with the non-inflamed phenotype were significantly associated with worse OS in the HER2-positive subtype (*p* < 0.001) and patients with the Cold T-cell infiltrated phenotype were significantly associated with worse OS in the TN subtype (*p* = 0.05). In the METABRIC cohort, patients in the hot T-cell infiltrated phenotype were marginally associated with better BCSS in whole cohort (*p* = 0.097), and patients with the Cold T-cell infiltrated phenotype were significantly associated with worse BCSS in the TN subtype (*p* = 0.009) (Figure 5). Xiao and colleagues reported that “immune-inflamed” cluster, with abundant adaptive and innate immune cells infiltration, had significantly better RFS and OS than the other two clusters in TNBC [14]. Interestingly, they demonstrated in the time-dependent AUC that the addition of microenvironment clusters into the Cox proportional hazards model significantly increased the prognostic efficacy of 1- and 2-year recurrence. Additionally, we demonstrated that, in distant recurrence analysis, patients with the hot T-cell infiltrated phenotype were significantly associated with better prognosis in whole cohort (*p* = 0.034) and the TN subtype (*p* = 0.019), but there was no significant correlation between our classification and local recurrence (Figure 5 and Appendix A). We previously reported that, in distant recurrence analysis, late recurrence was associated with activation of pro-cancerous immune cells. However in local recurrence analysis, there was no statistically significant difference in timing of cancer recurrences [5]. Inferring from the above results, immunofunctional phenotype was deeply involved not only in prognosis but also in the timing and type of recurrence of BC.

Finally, the infiltration of cytolytic CD8+ T cells in breast tumors was inversely related to estrogen response. In the analysis of the relationship between clinical features of the primary tumor and immunofunctional phenotype, patients with the hot T-cell infiltrated phenotype were significantly associated with TNBC or basal-like phenotype, compared with other groups (Table 1 and Table 2). In the pre-ranked GSEA, early and late estrogen responses were enriched in both the non-inflamed and Cold T-cell infiltrated phenotypes, compared with the hot T-cell infiltrated phenotype (Figure 4). These results confirmed that a recent bioinformatics study reporting increased immune infiltrate including cytotoxic T lymphocytes in ER-negative breast tumors relative to ER-positive breast tumors [15]. Further studies from the same group have shown that some of the ER-positive breast tumors treated with aromatase inhibitor had increased infiltration of B cell and T helper lymphocyte subsets, which may be predictive for effective BC treatment.

The methods for assessing TIME are very diverse and, due to these differences, individual studies cannot be compared to each other. Liquid biopsy is a method for extracting and analyzing oncogenes from body fluids such as blood and urine, and we have previously demonstrated the utility of BC-related genes in a non-invasive manner [16,17,18,19,20,21,22]. If TIME and its cytolytic activity can be monitored by liquid biopsy, this method is expected to better understand the true clinical and prognostic value of immune system cells in BC patients.

Although the study demonstrates promising results, it has limitations. First, this is a retrospective study utilizing publicly available datasets, thus some of the clinical factors deemed necessary were missing. It is necessary to verify the findings obtained with in-house clinical data. Second, this study is based on bioinformatics analysis of gene expression in primary breast tumor and does not include in vitro or in vivo experiments. Therefore, the mechanism for further understanding the results has not been clarified. Finally, there are issues with whether bulk genetics data from a tumor is enough to reveal relevant details regarding infiltration patterns. Additionally, some groups revealed gene expression differences between different tumor subtypes and their different TIMEs might lead to inaccuracies in the cell type separation [23,24]. In addition to these results, TCGA and METABRIC cohorts having very different clinical backgrounds may be causing the distributions of Tregs being exactly opposite. Further studies should perform to validate these bioinformatics approaches utilizing some data taken from more direct imaging studies of tissue samples.

## 4. Materials and Methods

### 4.1. Data Acquisition

In TCGA analysis, which was supervised by the National Cancer Institute (NCI) and the National Human Genome Research Institute [25], gene expression levels (mRNA expression z-score from RNA-sequence) were downloaded through cBioportal (TCGA PanCancer Atlas dataset) [26,27]. The phenotype, TMB, non-silent/silent mutation rate, SNV/Indel neoantigens, and duration of PFS/OS were obtained from (Liu et al., 2018 dataset) [28]. In METABRIC analysis, gene expression levels (mRNA expression z-score from microarray) were downloaded through cBioportal (METABRIC Nature 2012 & Nat Commun 2016 dataset). Additionally, the values of relapse phenotype (distant and local) and their relapse time were obtained from (Rueda et al., 2019 dataset) [29]. A total of 1075 women with BC in the TCGA cohort and a total of 1904 women with BC in the METABRIC cohort were used to support the authenticity of the association between immunofunctional phenotype and gene expression and TILs [30,31].

### 4.2. A CIBERSORT Deconvolution Algorithm

A cell fraction of 22 immune cells in each tumor tissue for assessing intra-tumor immune cell composition were obtained via the online calculator [32] based on CIBERSORT deconvolution algorithm [8], as previously shown [5,33,34].

### 4.3. The Immune Cytolytic Activity (CYT) Score

CYT was defined as the geometric mean of GZMA and PRF1 expression values in Transcripts Per Million (TPM). Those gene expression data were obtained in RSEM format from the Genomic Data Common data and converted to TPM by multiplying the estimated transcript of a particular gene by 1 × 10^6^ [10,35], as previously shown [5,33,34,36,37,38,39,40,41,42,43].

### 4.4. Statistical Analyses of RNA Expression and Gene Set Enrichment Analyses (GSEA)

Analysis followed a two-step process as previously described [5]. We first calculated the fold changes of genes, corresponding to immunofunctional phenotype (hot T-cell infiltrated, cold T-cell infiltrated, non-inflamed), which provided a list of *t*-scores and corresponding *p*-values for each immunofunctional phenotype in relation to each of the gene’s expression values. The second step is to run GSEA Pre-ranked utilizing the Hallmarks gene set using software provided by the Broad Institute [44]. We only considered gene sets significantly enriched that met a threshold of NES >1.7 or <−1.7 and FDR *q*-value < 0.01.

### 4.5. Statistical Analysis

All statistical analyses were performed using R software [45] and Bioconductor [46]. We evaluated the baseline differences between binary variables using the chi-square test, Fisher’s exact test, or the nonparametric Mann-Whitney U test and contingency analysis. The correlation was calculated using Spearman’s rank correlation coefficient. In the analysis of PFS, recurrence-free survival (RFS), OS, and breast cancer specific survival (BCSS), the Kaplan–Meier method was used to estimate survival rates, and differences between survival curves were evaluated by the log-rank test. Two-sided *p* values < 0.05 was considered as statistically significant for all tests.

## 5. Conclusions

We demonstrated the relationship between immunofunctional phenotype and clinical factors, TIME, molecular subtypes and survival utilizing collected data from the TCGA and METABRIC cohorts. We revealed that immunofunctional phenotype not only indicated the degree of anti-cancer immune dysfunction, but also served as a prognostic biomarker. ‘hot T-cell infiltrated’ tumors, with abundant cytolytic CD8+ T cells in breast tumors, were inversely related to estrogen response. Based on these reported results, we anticipate that further research can be conducted to establish a greater understanding of the role of TIME in BC.

## Figures and Tables

**Figure 1 cancers-13-00110-f001:**
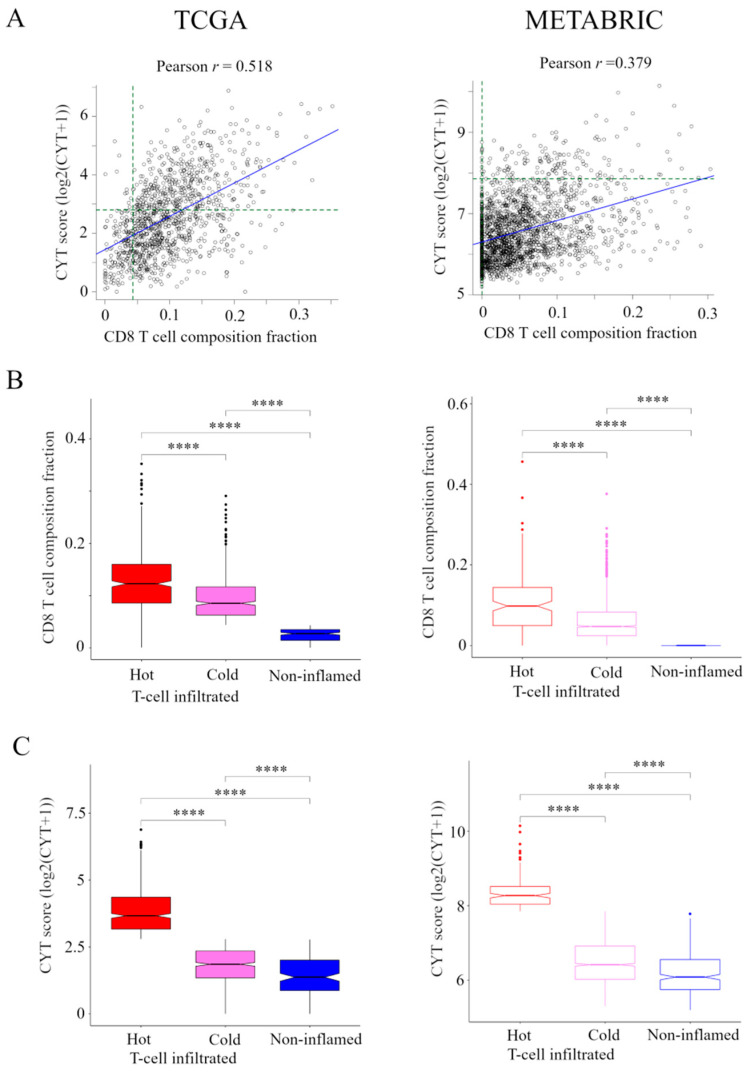
Classification of the immune phenotype based on the number and CYT of tumor-infiltrating CD8+ T cells in TCGA and METABRIC cohorts. (**A**) Correlation between the presence of CD8+ T cells and CYT in TCGA and METABRIC. We classified the immune phenotype into following three categories: hot T-cell infiltrated, a group of patients who had high CYT, Cold T-cell infiltrated, a group of patients who had high frequency of CD8+ T cells and low CYT, and Non-inflamed, a group of patients who had low frequency of CD8+ T cells and low CYT. For this classification, we defined the presence of CD8+ T cells of more than 85 th percentile of as high frequency of CD8+ T cells in TCGA and METABRIC cohorts and we defined CYT of more than 40 th percentile of as high CYT in the TCGA cohort and CYT of more than 8 th percentile of as high CYT in METABRIC cohort (green broken line). (**B**) and (**C**) The box plots of the presence of CD8+ T cells and CYT comparison between immunofunctional phenotype in TCGA BC cohort and METABRIC cohort. B, CD8+ T cells and C, CYT scores were shown. **** means *p* < 0.0001. Abbreviations: CIC, cancer immunity cycle; CYT, immune cytolytic activity; TCGA, The Cancer Genome Atlas; METABRIC, Molecular Taxonomy of Breast Cancer International Consortium.

**Figure 2 cancers-13-00110-f002:**
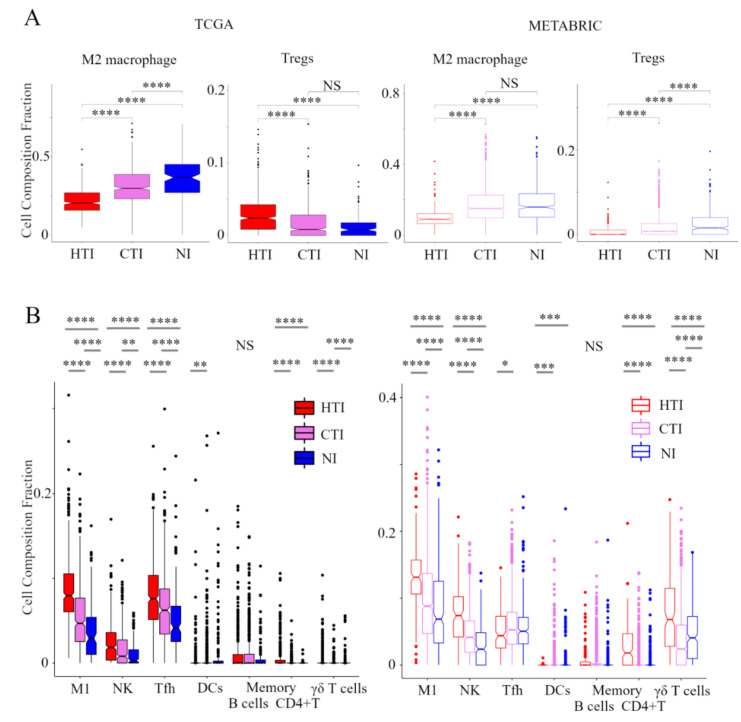
Verification of the relationship between the immunofunctional phenotype and immune cell fractions. Box plots of the relationship between each immunofunctional phenotype and immune cell fractions: pro-cancerous immune cells (M2 macrophage and Tregs); (**A**) and anti-cancer immune cells (left to right; M1 macrophage, activated NK cells, T follicular helper cells, memory B cells, activated DCs, memory B cells, activated memory CD4+ T cells, and γδT cells); (**B**) were shown. Immunofunctional phenotype was categorized as follows: hot T-cell infiltrated, a group of patients who had high CYT, cold T-cell infiltrated, a group of patients who had high frequency of CD8+ T cells and low CYT, and Non-inflamed, a group of patients who had low frequency of CD8+ T cells and low CYT. **** means *p* < 0.0001, *** means *p* < 0.001, ** means *p* < 0.01 and * means *p* < 0.05. Abbreviations: TCGA, The Cancer Genome Atlas; BC, METABRIC, Molecular Taxonomy of Breast Cancer International Consortium; HTI, hot T-cell infiltrated; CTI, cold T-cell infiltrated; NI, Non-inflamed; CIC, cancer immunity cycle; Tregs, CD4+ regulatory T cells; M1, M1 macrophage; NK, activated NK cells; Tfh, follicular helper cells; DCs, dendritic cells; CYT, immune cytolytic activity; NS, not significant.

**Figure 3 cancers-13-00110-f003:**
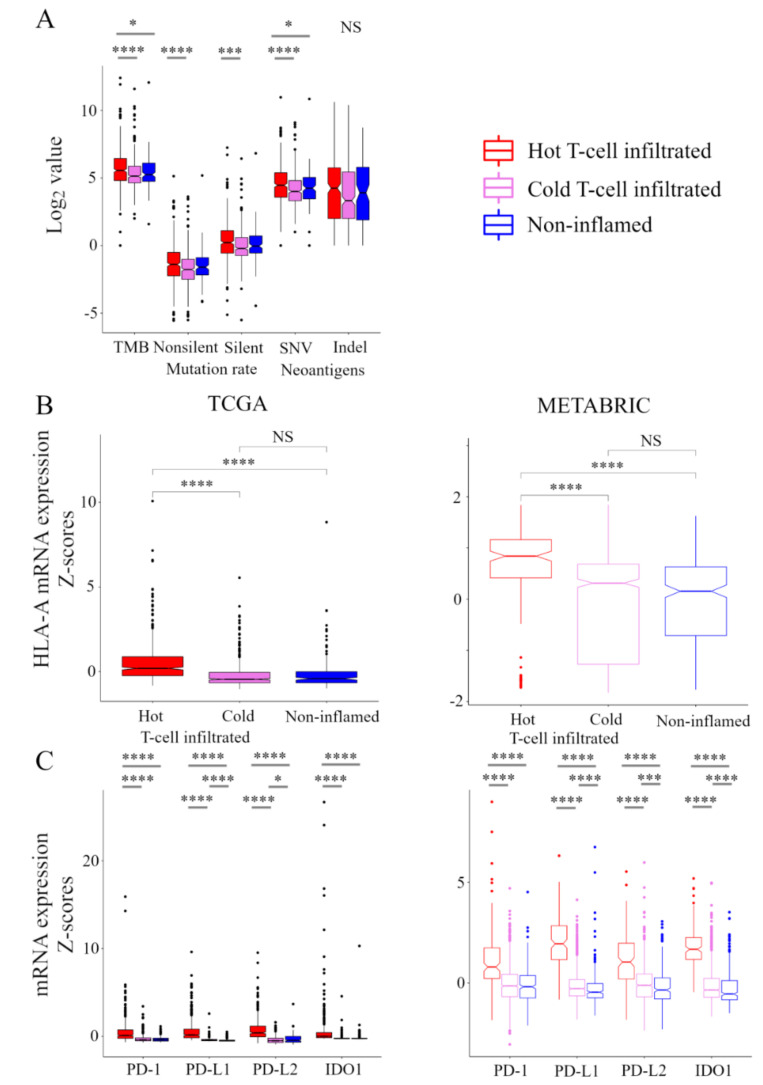
Stimulatory and inhibitory factors of the CIC in the immunofunctional phenotype. Box plots of the relationship between each immunofunctional phenotype and stimulatory or inhibitory factors of the CIC. (**A**) stimulatory factors of release of cancer cell antigens (left to right; TMB, non-silent mutation rate, silent mutation rate, SNV neoantigens, and Indel neoantigens), (**B**) stimulatory factor of recognition of cancer cells (HLA-A mRNA), and (**C**) killing of cancer cells (left to right; PD-1, PD-L1, PD-L2, and IDO1) were shown. Immunofunctional phenotype was categorized as follows: hot T-cell infiltrated, a group of patients who had high CYT, cold T-cell infiltrated, a group of patients who had high frequency of CD8+ T cells and low CYT, and Non-inflamed, a group of patients who had low frequency of CD8+ T cells and low CYT. **** means *p* < 0.0001, *** means *p* < 0.001, and * means *p* < 0.05. Abbreviations: CIC, cancer immunity cycle; TCGA, The Cancer Genome Atlas; METABRIC, Molecular Taxonomy of Breast Cancer International Consortium; TMB, tumor mutation burden; SNV, single nucleotide variant; Indel, insertion/deletion; CYT, immune cytolytic activity; NS, not significant.

**Figure 4 cancers-13-00110-f004:**
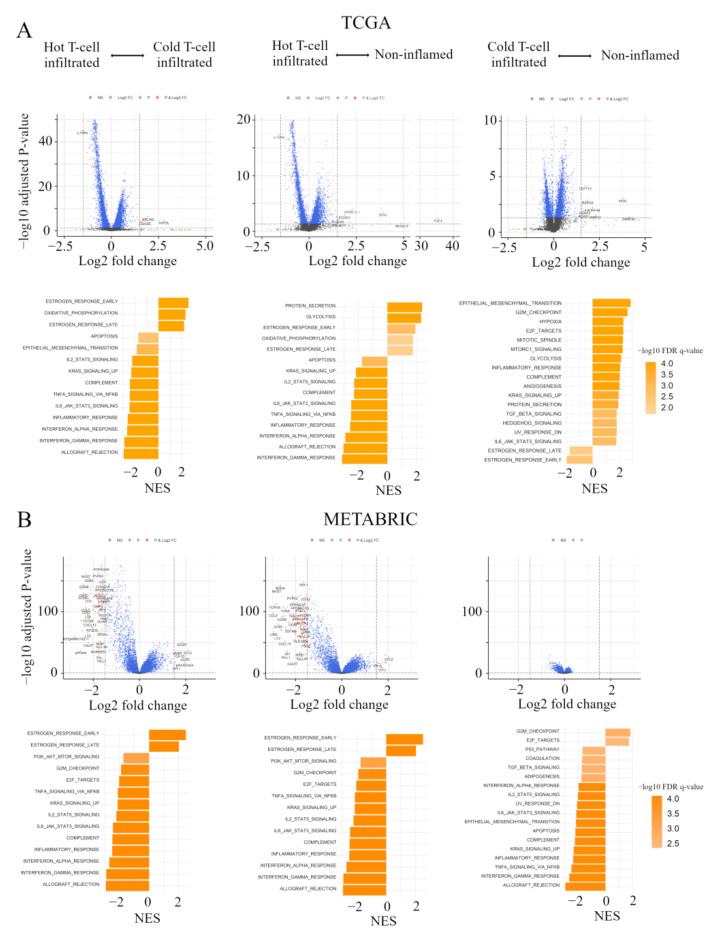
Gene expression profiles in the immunofunctional phenotype. (**A**) Volcano plots illustrating the differentially expressed mRNAs of BC and pre-ranked GSEA of BC patients comparing hot T-cell infiltrated vs cold T-cell infiltrated, hot T-cell infiltrated vs Non-inflamed, and cold T-cell infiltrated vs non-inflamed in TCGA BC cohort; A and METABRIC cohort; (**B**) Upper panels: In volcano plots, *X*-axes: log2 FC; Y-axes: -log 10 *p*-value from limma analysis. mRNAs with *p*-value < 0.05 and FC > 1.5 are marked in red, with *p*-value < 0.05 and FC < 1/1.5 in green, all others in black. Bottom panels: In pre-ranked GSEA, orange bar shows NES and the shading of it show –log10 FDR q-value. We only considered gene sets significantly enriched that met a threshold of NES >1.7 or <−1.7 and FDR q-value < 0.01. Immunofunctional phenotype was categorized as follows: hot T-cell infiltrated, a group of patients who had high CYT, cold T-cell infiltrated, a group of patients who had high frequency of CD8+ T cells and low CYT, and non-inflamed, a group of patients who had low frequency of CD8+ T cells and low CYT. Abbreviations: CIC, cancer immunity cycle; BC, breast cancer; GSEA, Gene Set Enrichment Analyses; TCGA, The Cancer Genome Atlas; METABRIC, Molecular Taxonomy of Breast Cancer International Consortium; FC, fold change; NES, normalized enrichment score; FDR, false discovery rate; CYT, immune cytolytic activity.

**Figure 5 cancers-13-00110-f005:**
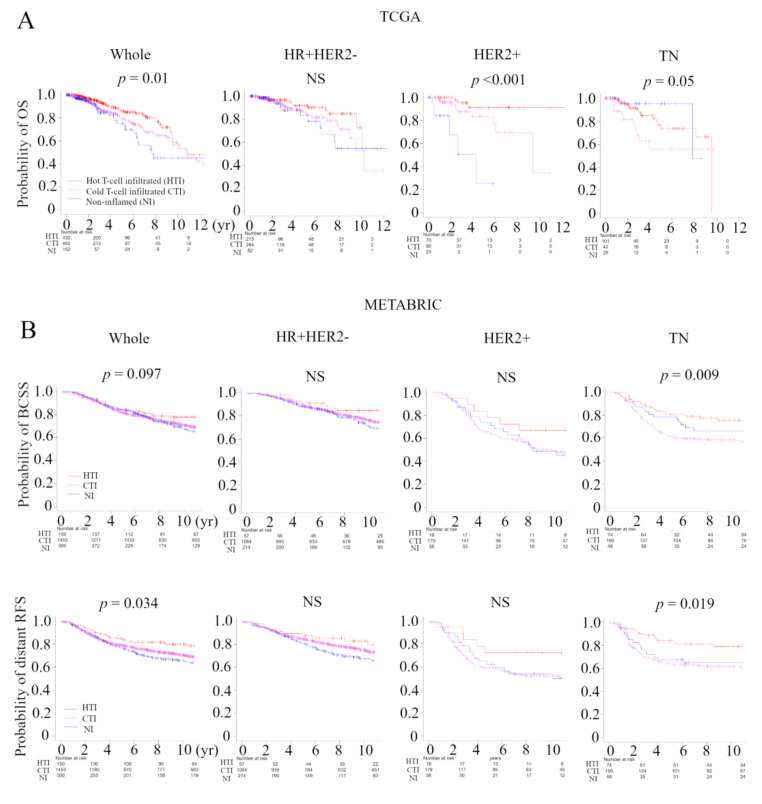
Analysis of the relationship between the immunofunctional phenotype and survival in TCGA and METABRIC cohorts. (**A**) Kaplan-Meier plots of the association of the immunofunctional phenotype with OS in TCGA; A and BCSS and distant RFS in METABRIC; (**B**) (left to right) the whole cohort, the HR+HER2- group, the HER2+ group, and the TN group. Immunofunctional phenotype was categorized as follows: hot T-cell infiltrated, a group of patients who had high CYT, Cold T-cell infiltrated, a group of patients who had high frequency of CD8+ T cells and low CYT, and non-inflamed, a group of patients who had low frequency of CD8+ T cells and low CYT. Abbreviations: CIC, cancer immunity cycle; TCGA, The Cancer Genome Atlas; METABRIC, Molecular Taxonomy of Breast Cancer International Consortium; OS, overall survival; BCSS, breast cancer specific survival; RFS, recurrent-free survival; HR, hormone receptor; HER2, human epidermal growth factor receptor 2; TN, triple negative; CYT, immune cytolytic activity; NS, not significant.

**Table 1 cancers-13-00110-t001:** Patients and clinical characteristics associated with the immunofunctional phenotype in TCGA cohort.

Variables		Number of Patients (%)				
		Total	Immunofunctional Phenotype			
			Hot T-Cell Infiltrated	Cold T-Cell Infiltrated	Non-Inflamed	*p* Value
		(*n* = 1075)	(*n* = 431)	(*n* = 492)	(*n* = 152)	
Age	50≥	291 (27.1)	130 (30.2)	115 (23.4)	46 (30.3)	0.043 *
	50<	784 (72.9)	301 (69.8)	377 (76.6)	106 (69.7)	
Race	Caucasian American	742 (69)	297 (68.9)	333 (67.7)	112 (73.7)	0.75
	African American	182 (16.9)	72 (16.7)	87 (17.7)	23 (15.1)	
	Asian	61 (5.7)	28 (6.5)	26 (5.3)	7 (4.6)	
	Unknown	90 (8.4)	34 (7.9)	46 (9.3)	10 (6.6)	
Menopausal state	Pre	224 (20.8)	97 (22.5)	94 (19.1)	33 (21.7)	0.66
	Post	691 (64.3)	282 (65.4)	314 (63.8)	95 (62.5)	
	Unknown	160 (14.9)	52 (12.1)	84 (17.1)	24 (15.8)	
Tumor size (cm)	2≥	898 (83.5)	365 (84.7)	399 (81.1)	134 (88.2)	0.11
	2<	175 (16.3)	66 (15.3)	91 (18.5)	18 (11.8)	
	Unknown	2 (0.2)	0	2 (0.4)	0	
Lymphnode	Negative	509 (47.3)	201 (46.6)	224 (45.5)	84 (55.3)	0.08
	Positive	547 (50.9)	227 (52.7)	256 (52)	64 (42.1)	
	Unknown	19 (1.8)	3 (0.7)	12 (2.4)	4 (2.6)	
Histopathology	Ductal	775 (72.1)	290 (67.3)	360 (73.2)	125 (82.2)	<0.0001 *
	Lobular	199 (18.5)	109 (25.3)	81 (16.5)	9 (5.9)	
	Others	101 (9.4)	32 (7.4)	51 (10.4)	18 (11.8)	
Clinical stage	I/II	787 (73.2)	312 (72.4)	355 (72.2)	120 (78.9)	0.28
	III/IV	265 (24.7)	109 (25.3)	126 (25.6)	30 (19.7)	
	Unknown	23 (2.1)	10 (2.3)	11 (2.2)	2 (1.3)	
ER	Negative	236 (22)	129 (29.9)	68 (13.8)	39 (25.7)	<0.0001 *
	Positive	789 (73.4)	287 (66.6)	397 (80.7)	105 (69.1)	
	Unknown	50 (4.7)	15 (3.5)	27 (5.5)	8 (5.3)	
PgR	Negative	340 (31.6)	158 (36.7)	133 (27)	49 (32.2)	0.011 *
	Positive	686 (63.8)	258 (59.9)	333 (67.7)	95 (62.5)	
	unknown	49 (4.6)	15 (3.5)	26 (5.3)	8 (5.3)	
HER2	Negative	754 (70.1)	317 (73.5)	326 (66.3)	111 (73)	0.38
	Positive	182 (16.9)	70 (16.2)	89 (18.1)	23 (15.1)	
	Unknown	139 (12.9)	44 (10.2)	77 (15.7)	18 (11.8)	
Subtype	HR+ ^a^ HER2-	582 (54.1)	216 (50.1)	284 (57.7)	82 (53.9)	<0.0001 *
	HER2+	181 (16.8)	70 (16.2)	88 (17.9)	23 (15.1)	
	TN ^b^	172 (16)	101 (23.4)	42 (8.5)	29 (19.1)	
	Unknown	140 (13)	44 (10.2)	78 (15.9)	18 (11.8)	
PAM50	Luminal A	413 (38.4)	141 (32.7)	211 (42.9)	61 (40.1)	<0.0001 *
	Luminal B	188 (17.5)	57 (13.2)	96 (19.5)	35 (23)	
	HER2	67 (6.2)	33 (7.7)	28 (5.7)	6 (3.9)	
	Basal-like	138 (12.8)	76 (17.6)	40 (8.1)	22 (14.5)	
	Normal	23 (2.1)	15 (3.5)	6 (1.2)	2 (1.3)	
	Unknown	246 (22.9)	109 (25.3)	111 (22.6)	26 (17.1)	

Abbreviations: TCGA, The Cancer Genome Atlas; ER, estrogen receptor; PgR, progesterone receptor; HER2, human epidermal growth factor receptor 2; HR, hormone receptor; TN, triple negative. ^a^ HR+: ER-positive and/or PgR-positive. ^b^ TN: HR-negative and HER2-negative. * Factor showing statistical significance. The chi-square test and Fisher’s extract test were used to assess baseline differences between binary variables. *p* < 0.05 is considered statistically significant.

**Table 2 cancers-13-00110-t002:** Patients and clinical characteristics associated with the immunofunctional phenotype in METABRIC cohort.

Variables		Number of Patients (%)				
		Total	Immunofunctional Phenotype			
			Hot T-Cell Infiltrated	Cold T-Cell Infiltrated	Non-Inflamed	*p* Value
		(*n* = 1904)	(*n* = 150)	(*n* = 1454)	(*n* = 300)	
Age	50≥	411 (21.6)	43 (28.7)	306 (21)	62 (20.7)	0.088
	50<	1493 (78.4)	107 (71.3)	1148 (79)	238 (79.3)	
Menopausal state	Pre	411 (21.6)	43 (28.7)	306 (21)	62 (20.7)	0.088
	Post	1493 (78.4)	107 (71.3)	1148 (79)	238 (79.3)	
Tumor size (cm)	2≥	821 (43.1)	71 (47.3)	630 (43.3)	120 (40)	0.28
	2<	1063 (55.8)	77 (51.3)	808 (55.6)	178 (59.3)	
	Unknown	20 (1.1)	2 (1.3)	16 (1.1)	2 (0.7)	
Lymphnode	Negative	993 (52.2)	81 (54)	754 (51.9)	158 (52.7)	0.87
	Positive	911 (47.8)	69 (46)	700 (48.1)	142 (47.3)	
Histopathology	Ductal	1454 (76.4)	106 (70.7)	1107 (76.1)	241 (80.3)	0.19
	Lobular	142 (7.5)	12 (8)	113 (7.8)	17 (5.7)	
	Others/unknown	308 (16.2)	32 (21.3)	234 (16.1)	42 (14)	
Tumor grade	1/2	905 (47.5)	42 (28)	720 (49.5)	143 (47.7)	<0.0001 *
	3	927 (48.7)	106 (70.7)	674 (46.4)	147 (49)	
	unknown	72 (3.8)	2 (1.3)	60 (4.1)	10 (3.3)	
Clinical Stage	I/II	1275 (67)	102 (68)	961 (66.1)	212 (70.7)	0.16
	III/IV	124 (6.5)	15 (10)	94 (6.5)	15 (5)	
	Unknown	505 (26.5)	33 (22)	399 (27.4)	73 (24.3)	
ER	Negative	445 (23.4)	85 (56.7)	288 (19.8)	72 (24)	<0.0001 *
	Positive	1459 (76.6)	65 (43.3)	1166 (80.2)	228 (76)	
PgR	Negative	895 (47)	122 (81.3)	641 (44.1)	132 (44)	<0.0001 *
	Positive	1009 (53)	28 (18.7)	813 (55.9)	168 (56)	
HER2	Negative	1668 (87.6)	131 (87.3)	1275 (87.7)	262 (87.3)	0.98
	Positive	236 (12.4)	19 (12.7)	179 (12.3)	38 (12.7)	
Subtype	HR+ ^a^ HER2-	1355 (71.2)	57 (38)	1084 (74.6)	214 (71.3)	<0.0001 *
	HER2+	236 (12.4)	19 (12.7)	179 (12.3)	38 (12.7)	
	TN ^b^	312 (16.4)	74 (49.3)	190 (13.1)	48 (16)	
	Unknown	1 (0.1)	0	1 (0.1)	0	
Molecular Characterization	Luminal A	679 (35.7)	13 (8.7)	564 (38.8)	102 (34)	<0.0001 *
	Luminal B	461 (24.2)	14 (9.3)	360 (24.8)	87 (29)	
	HER2	220 (11.6)	18 (12)	158 (10.9)	44 (14.7)	
	Basal-like	199 (10.5)	13 (8.7)	150 (10.3)	36 (12)	
	Claudin-low	199 (10.5)	89 (59.3)	96 (6.6)	14 (4.7)	
	Normal	140 (7.4)	3 (2)	122 (8.4)	15 (5)	
	Unknown	6 (0.3)	0	4 (0.3)	2 (0.7)	

Abbreviations: METABRIC, Molecular Taxonomy of Breast Cancer International Consortium; ER, estrogen receptor; PgR, progesterone receptor; HER2, human epidermal growth factor receptor 2; HR, hormone receptor; TN, triple negative. ^a^ HR+: ER-positive and/or PgR-positive. ^b^ TN: HR-negative and HER2-negative. * Factor showing statistical significance. The chi-square test and Fisher’s extract test were used to assess baseline differences between binary variables. *p* < 0.05 is considered statistically significant.

## Data Availability

The data presented in this study are available on request from the corresponding author.

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
