# Peer review of "The Impact of Immunofunctional Phenotyping on the Malfunction of the Cancer Immunity Cycle in Breast Cancer"

_cancers, 2020, doi:10.3390/cancers13010110_

Round 1

Reviewer 1 Report

Overall design and flow of study is quite excellent and well suited. Proper sorting of TIME and adequate validation are also remarkable. 

I think it cold be acceptable as its present form. 

Author Response

Comment 1)

Overall design and flow of study is quite excellent and well suited. Proper sorting of TIME and adequate validation are also remarkable. 

I think it could be acceptable as its present form. 

Response 1)

Thank you so much.

Reviewer 2 Report

The manuscript entitled “Impact of immunofunctional phenotyping the malfunction of the cancer immunity cycle in breast cancer” by Takeshita et al. is a very interesting study where authors have demonstrated the correlation between multiple aspects of breast cancer. They have established the relationship between cellular immune profiles, tumor microenvironment, estrogen responsiveness and survival/prognosis in various clinical samples obtained from women with breast cancer. This study can provide useful insights to future clinical studies.

Author Response

Comment 1)

The manuscript entitled “Impact of immunofunctional phenotyping the malfunction of the cancer immunity cycle in breast cancer” by Takeshita et al. is a very interesting study where authors have demonstrated the correlation between multiple aspects of breast cancer. They have established the relationship between cellular immune profiles, tumor microenvironment, estrogen responsiveness and survival/prognosis in various clinical samples obtained from women with breast cancer. This study can provide useful insights to future clinical studies.

Response 1)

Thank you so much.

Reviewer 3 Report

This paper presents the results of purely bioinformatic analyses of the state of immune response in public databases of breast cancer patients. While the results are potentially interesting, I found a lack of any critical discussion of the methods being used and the data being analyzed. This is a very common failing of papers such as this one, but nonetheless the authors need to try harder. Specifically,

  • There are clear differences in the results from the two different data sources. For example, the distributions of regulatory t-cells are exactly opposite, even though each one is claimed to have extremely high statistical significance. This leaves two possibilities – there are systematic differences between the two datasets or the p-value calculations are just plain wrong. The authors note the differences and then just move on; can they at least present some serious discussion of what could be going on? The same difficulty always appears when lists of relevant genes and/or pathways determined by fold change differ from dataset to dataset.
  • The authors have placed all their trust in programs like Cibersort to disentangle gene expressions without even considering the possibility that gene expression differences between different tumor subtypes and their different TIMEs might lead to inaccuracies in the cell type separation. There are also issues with whether bulk genetics data from a tumor is enough to reveal relevant details regarding infiltration patterns. The only way to validate these bioinformatics approaches is to consider at least some data taken from more direct imaging studies of tissue samples. Surprisingly the authors have not bothered to reference any of these studies Infiltration of CD8+ T cells into tumor cell clusters in triple-negative breast cancer. (Keren et al., 2018, Cell 174, 1373–1387; Li, et al, 2019, PNAS116 (9) 3678) or in other tumors and discuss the extent to which they agree or disagree with any of the findings here.
  • I don’t understand the significance bars in figure 2; the data was separated according to its CYT score and then significance was claimed for the fact that the separate categories had different CYT scores?? Am I missing something here.
  • The reference numbering is off. For example, the citation of reference [5] right at the end of the first paragraph in the main text is clearly meant to refer to [6]

Author Response

This paper presents the results of purely bioinformatic analyses of the state of immune response in public databases of breast cancer patients. While the results are potentially interesting, I found a lack of any critical discussion of the methods being used and the data being analyzed. This is a very common failing of papers such as this one, but nonetheless the authors need to try harder.

Comment 1)

Specifically, there are clear differences in the results from the two different data sources. For example, the distributions of regulatory t-cells are exactly opposite, even though each one is claimed to have extremely high statistical significance. This leaves two possibilities – there are systematic differences between the two datasets or the p-value calculations are just plain wrong. The authors note the differences and then just move on; can they at least present some serious discussion of what could be going on? The same difficulty always appears when lists of relevant genes and/or pathways determined by fold change differ from dataset to dataset.

The authors have placed all their trust in programs like Cibersort to disentangle gene expressions without even considering the possibility that gene expression differences between different tumor subtypes and their different TIMEs might lead to inaccuracies in the cell type separation.

There are also issues with whether bulk genetics data from a tumor is enough to reveal relevant details regarding infiltration patterns. The only way to validate these bioinformatics approaches is to consider at least some data taken from more direct imaging studies of tissue samples. Surprisingly the authors have not bothered to reference any of these studies Infiltration of CD8+ T cells into tumor cell clusters in triple-negative breast cancer. (Keren et al., 2018, Cell 174, 1373–1387; Li, et al, 2019, PNAS116 (9) 3678) or in other tumors and discuss the extent to which they agree or disagree with any of the findings here.

Response 1)

Thank you for your suggestion. As you mentioned, the distributions of regulatory t-cells are exactly opposite, even though each one is claimed to have extremely high statistical significance. Furthers, we agree that the programs we used in this study have the possibility that gene expression differences between different tumor subtypes and their different TIMEs might lead to inaccuracies in the cell type separation. To make our argument clearer, we added the following paragraph to the limitation part in "Discussion".

“Finally, there are issues with whether bulk genetics data from a tumor is enough to reveal relevant details regarding infiltration patterns. Additionally, some groups revealed gene expression differences between different tumor subtypes and their different TIMEs might lead to inaccuracies in the cell type separation [23,24]. In addition to these results, TCGA and METABRIC cohorts having very different clinical backgrounds may be causing the distributions of regulatory T-cells being exactly opposite, even though each one is claimed to have extremely high statistical significance Further studies should perform to validate these bioinformatics approaches utilizing some data taken from more direct imaging studies of tissue samples.”

Reviewer 4 Report

This is secondary analysis of the open source data sets. This is completely a bioinformatics approach. A minor drawback is there are no functional studies performed on the high target genes, which would be out of scope of this article. However, that being said, this is an extremely well-written and thorough manuscript and should be published. 

I have no concerns. 

Author Response

Comment 1)

This is secondary analysis of the open source data sets. This is completely a bioinformatics approach. A minor drawback is there are no functional studies performed on the high target genes, which would be out of scope of this article. However, that being said, this is an extremely well-written and thorough manuscript and should be published. 

I have no concerns. 

Response 1)

Thank you so much.

Round 2

Reviewer 3 Report

I now support publication of this work. The authors have responded, albeit in the most minimalist fashion, to my concerns and this paper is no better or worse than most other published papers using these methods.